# HiBug: On Human-Interpretable Model Debug

**Muxi Chen[†][*], Yu Li[‡][*], Qiang Xu[†]**

[†] The Chinese University of Hong Kong
[‡] Harbin Institute of Technology, Shenzhen
{mxchen21,qxu}@cse.cuhk.edu.hk; li.yu@hit.edu.cn

## Abstract

Machine learning models can frequently produce systematic errors on critical subsets (or slices) of data that share common attributes. Discovering and explaining such model bugs is crucial for reliable model deployment. However, existing bug discovery and interpretation methods usually involve heavy human intervention and annotation, which can be cumbersome and have low bug coverage.

In this paper, we propose *HiBug*, an automated framework for interpretable model debugging. Our approach utilizes large pre-trained models, such as chatGPT, to suggest human-understandable attributes that are related to the targeted computer vision tasks. By leveraging pre-trained vision-language models, we can efficiently identify common visual attributes of underperforming data slices using human-understandable terms. This enables us to uncover rare cases in the training data, identify spurious correlations in the model, and use the interpretable debug results to select or generate new training data for model improvement. Experimental results demonstrate the efficacy of the HiBug framework. Code is available at: https://github.com/cure-lab/HiBug.

## 1 Introduction

Machine learning models for computer vision tasks have made significant advances in recent years. However, they are still prone to systematic failures on critical subsets, or "slices," of data with similar visual attributes. These bugs are largely due to the dataset used for training, such as rare data in certain categories or biased training data leading to spurious correlations, causing potentially catastrophic consequences for models used in safety-critical applications, such as autonomous driving [Fujiyoshi et al., 2019] and medical image processing [Giger, 2018].

Various model debugging techniques are proposed in the literature. Some focus on discovering failure instances. For example, test input prioritization techniques aim to identify as many failure instances as possible under a given labeling budget [Li et al., 2021, Wang et al., 2021]. Other solutions seek to identify the visual attributes that cause model failures. For example, [Singla et al., 2021] proposes a visualization technique (e.g., saliency maps), through which humans can identify bug-related visual attributes. The recent Domino [Eyuboglu et al., 2022] and Spotlight [d'Eon et al., 2022] techniques try to cluster failures in the embedding space for common bug attribute identification. While interesting, the visual attributes in the embedding space are often entangled, making it hard to identify the true bug-related attributes. Adavision [Gao et al., 2022] proposes to describe model bugs with natural languages and refine the description interactively. Such a method requires heavy human intervention and cannot guarantee bug coverage.

In this paper, we propose *HiBug*, an automated framework for interpretable model debugging. HiBug leverages the recent large pre-trained models to identify common visual attributes of underperforming

---

[*]These authors contributed equally.

37th Conference on Neural Information Processing Systems (NeurIPS 2023).

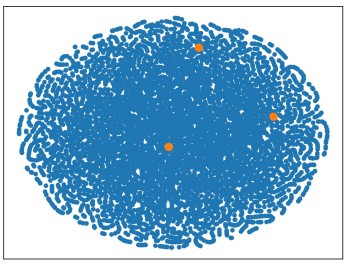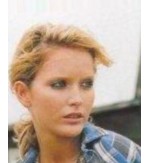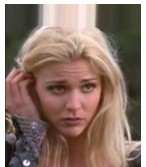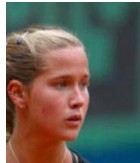

Gender: Female
Age: Young
Hair Color: Blonde
Skin: White
Emotion: Serious
Beard: No
Makeup: No

Figure 1: Visualization of the CLIP's embedding space. As we can observe, the visually similar images are distributed across the embedding space (yellow dots) . This is because the embedding space is entangled and hence cannot accurately reflect visual similarity.

data slices. Specifically, given the targeted computer vision task, we first employ a pre-trained large language model such as chatGPT to collect a corpus of visual attributes that are related to this task. Next, we attach these attributes to all the images, leveraging pre-trained vision-language models such as BLIP. Afterwards, we perform data slicing based on these attributes and conduct statistical analysis to identify those underperforming slices with common visual attributes. Such a bug identification procedure is naturally human-interpretable. We further conduct failure reasoning and use the debug results to select or generate training data for model repair.

To sum up, the main contributions of this paper include:

- HiBug is a highly automated and interpretable model debugging framework. It achieves this by introducing a new debugging flow to identify failure instances and describe the visual attributes causing failures using natural language.
- We propose to leverage the capability of the powerful pre-trained models for interpretable model debugging, which efficiently identifies common visual attributes of underperforming data slices using human-understandable terms.
- Based on the discovered failure attributes, we further conduct failure reasoning, i.e., classifying if the failure is caused by spurious correlation or lack of training data. Moreover, we show that our framework can be directly used to guide the selection and generation of more images with failure attributes, which facilitate model repair.

Our experiments on dcbench [Eyuboglu et al., 2022] show that HiBug can identify up to 85% correlation errors and 81% rare cases. We also use three different tasks to show that HiBug benefits data selection and data generation for model improvement. The rest of this paper is organized as follows. Section 2 presents the related work. Section 3 illustrates our method and Section 4 shows the effectiveness of our method. Section 5 concludes this paper.

## 2 Related Works

**Test input prioritization.** Test input prioritization techniques are representative model bug discovery methods [Feng et al., 2020, Kim et al., 2019, Li et al., 2021]. Specifically, they aim to identify "high-quality" test instances from a large amount of unlabeled data, which can reveal more model failures with less labeling effort. DeepGini, as proposed by [Feng et al., 2020], rank unlabeled test instances by calculating confidence scores based on the model's output probabilities. Test cases that exhibit nearly equal probabilities across all output classes are considered less confident and are thus more likely to uncover model failures. Follow-up methods include considering the diversity of the selected instances or proposing more accurate metrics to evaluate the failure probability for one test instance. Despite the effectiveness in reducing the debug cost, these methods do not provide further explanation on why this failure happens.

**Interpretable bug discovery.** Instead of discovering more failures, interpretable bug discovery methods aim to find human-understandable visual attributes that cause the failures.

To this end, [Singla et al., 2021] present a sample-specific visualization technique based on the embedding space of the deep learning model. Specifically, for each failure sample, this method can

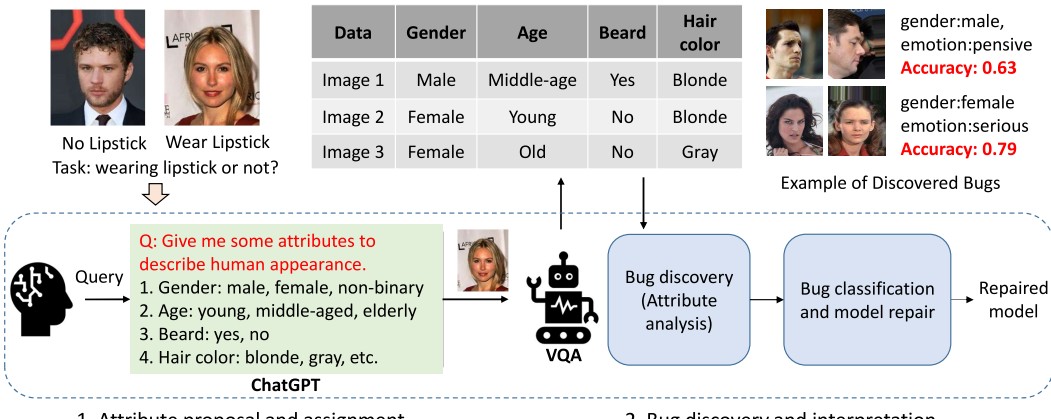

Figure 2: The overview of HiBug.

visualize the features that the model sees when making the decision, and thus provide some extent of interpretability. Spotlight [d'Eon et al., 2022] proposes to find the high failure rate area in the model's embedding space. The failures in that area exhibit some visual similarity. AdaVision [Gao et al., 2022] proposes a human-in-the-loop testing procedure. AdaVision first generates a set of potential captions with the help of the user and language model. Then, it matches data to the captions and iteratively selects captions with a high error rate in data. Next, it asks the user to add more details to refine the captions and continue the iteration. However, these methods involve a large amount of human effort. The work proposed by [Singla et al., 2021] and [d'Eon et al., 2022] requires humans to summarize the visual features that cause the failures, and AdaVision requires humans to refine the caption. Such heavy human intervention procedures are cumbersome and cannot guarantee high bug coverage.

Another line of work proposes to automatically summarize failure-related visual attributes using natural language. They cluster failure by mixture models [Eyuboglu et al., 2022] or a linear classifier [Radford et al., 2021]. They then generate a set of candidate captions by combining words and templates from a large, predefined corpus. Finally, they use CLIP [Radford et al., 2021] to match each data group with a caption.

However, the CLIP's embedding space often contains entangled visual attributes. As shown in Figure 1, this can result in semantically coherent images being far away in the embedding space. Furthermore, the candidate captions generated from the corpus might contain irrelevant or contradicting words, such as "a photo of setup by banana" and "a photo of skiing at sandal" [Gao et al., 2022]. Meanwhile, the corpus of candidate words is manually designed and thus requires intensive human effort. Finally, the generated captions only describe the property of failure data, failing to imply why the model goes wrong, i.e., because of rare training cases or suspicious dependence.

HiBug focuses on interpretable bug discovery and addresses the above challenges in several ways. First, unlike existing efforts that seek to cluster failures, HiBug follows a different debugging flow. We first group the data according to their attributes and then judge if there are underperforming ones. This way, underperforming slices must share a similar concept, leading to more accurate explanations. Second, HiBug eliminates the need for a manual-intensive external corpus by leveraging state-of-the-art large language models to propose task-specific attributes based on user-provided task descriptions. Lastly, HiBug supports the root cause discovery of the failures.

## 3 Methods

The overview of HiBug is presented in Figure 2. HiBug aims to systematically identify and interpret consistent bugs in trained models. As an illustrative example, we consider a model that classifies instances of lipstick usage. HiBug initiates this process by employing large, pre-existing language models, such as ChatGPT, to propose visual attributes pertinent to the task under investigation. Subsequently, for each image in the dataset, corresponding attribute values are assigned by leveraging vision-language models. We consider this step as the attribute proposal and assignment step. Upon

establishing these attributes for each image, HiBug can detect low-performing data slices, and provide natural language descriptions and root causes for the identified bugs. We consider this step as the bug discovery and interpretation step.

In the context of the lipstick-use classification model, HiBug might uncover a correlation between the model's predictions and gender, i.e., predict 'lipstick usage' for images depicting females. A retrospective examination of the issue reveals that the training dataset exhibits an inherent bias, with females frequently appearing with lipsticks. This results in the model developing spurious correlations, which in turn lead to erroneous predictions.

Lastly, HiBug demonstrates its efficacy by selectively generating or selecting novel data for retraining the model, thereby enhancing its overall performance. This approach facilitates the rectification of identified spurious correlations and biases that emerged during the bug discovery phase.

Please note that some components in HiBug, such as ChatGPT and BLIP, are selected mainly because they are easy to use and provide good solutions. HiBug is highly flexible and modular. Therefore, designers can update any component in HiBug with a better counterpart, if needed.

### 3.1 Attribute proposal and assignment

Images are distinguished by unique visual features. For example, an image might be described as 'A young woman with brown hair'. In this context, 'young', 'woman', and 'brown hair' constitute elements of visual semantics. However, these features aren't inherently known for each image, complicating the task of identifying interpretable bugs.

To articulate model bugs in a form understandable to humans, such as natural language, it is essential to establish a set of task-related visual attributes, denoted as $A$. These attributes serve to characterize images within a specific classification task. Using the previous example, task attributes, $A$, might include 'gender', 'age', and 'hair color'. Each attribute, $a \in A$, can assume several potential values, represented as $V(a)$. For instance, the attribute 'gender' could include values such as 'male', 'female', and 'non-binary'.

Hence, with a pre-defined set of attributes, $A$, and their corresponding values, $V(a)$, characterizing an image becomes a more structured process. However, determining these visual attributes and corresponding values is challenging. This difficulty stems from the absence of inherent knowledge about each image's visual features, which, if known in advance, would simplify the identification of interpretable bugs.

**Attribute name proposal.** Recent research indicates that large language models, like ChatGPT, possess extensive general knowledge and are highly effective at answering questions [Wei et al., 2022, Touvron et al., 2023]. We aim to utilize these capabilities to propose task-related visual attributes.

In this process, users are required to provide ChatGPT with a simple task description. For instance, in the lipstick-usage problem where the task is to identify lipstick on a human face, we pose the following question to ChatGPT: 'Can you list some attributes to describe human appearance?' ChatGPT then returns a list of potential attributes. An example of such a question-and-answer interaction with ChatGPT is illustrated in Figure 2. Upon receiving the responses, we compile each attribute $a \in A$ to form an attribute list $A$. It should be noted that the question should be task-specific, necessitating user input for its design. However, barring this step, the entire process is fully automated.

**Attribute value proposal.** Upon generating the attribute list $A$ for each task, we face the challenge of identifying corresponding attribute values. Although ChatGPT can offer some suggestions, its limited access to dataset images may lead to mismatches with the actual dataset distribution. To address this, we utilize a visual question-answering (VQA) model, which augments attribute value accuracy by analyzing dataset images and providing corresponding attribute value responses.

A straightforward approach to accumulating attribute values using the VQA model would involve posing a query for each image in the form of:

$$\text{"What is the *1 of the *2 in the image?"} \tag{1}$$

In this equation, *1 and *2 represent attributes and task-related elements collected previously. The VQA model's response serves as the attribute value for the image. However, this approach proved time-consuming due to the volume of images.

Therefore, we propose a more efficient approach: selecting representative images from the dataset for attribute value proposal. The rationale is that not every image needs to be queried for their attribute value types, as similar images tend to share similar attribute value types. For example, attribute values like female', male', and 'non-binary' can be gathered by querying just a few images. Querying additional images would likely not yield more diverse choices, but would merely consume more time. Hence, we can use representative images for attribute value proposals to reduce redundancy.

To select these representative images, we employ the pre-trained vision-language model, BLIP [Li et al., 2022], for feature extraction. The representative samples are selected by clustering the features. Then we query the VQA model (BLIP) with the representative samples for attribute value proposals. Finally, we aggregate all attribute values to form a value choice list $A(a)$ for each attribute name $a \in A$.

**Attribute assignment.** Our process begins with a set of attribute names, denoted as *A*, with each attribute name $a \in A$ having corresponding attribute values represented as *A(a)*. The goal is to assign an appropriate attribute value from *A(a)* for each attribute $a \in A$ to every image $x$.

To this end, we consider value from *A(a)* as labels and apply zero-shot classification on images with the BLIP model. The end result is the formation of an attribute matrix $X_A$, an illustrative example of which can be found in Figure 2.

## 3.2 Interpretable bug discovery and classification

**Interpretable bug discovery.** After the attribute matrix $X_A$ is obtained, we conduct bug discovery based on the matrix. The easiest way is to analyze the attributes separately (e.g., analysis if the 'young' attribute value lead to under-performing slices). Assuming the number of attributes is $m = |A|$, and the number of choices for each attribute is $n_i, i \in \{1..m\}$, then the total number of attribute choices is $N = \sum n_i, i \in \{1..m\}$. For each attribute choice, we group images with similar attributes and test their accuracy. In total, we can have $N$ groups and $N$ test accuracies.

To take a step further, we can analyze the slices defined by combinations of the attributes (e.g., analysis if the 'young woman with brown hair' attribute value combinations leads to under-performing slices). Specifically, we group data with the same attribute value combinations into one data slice. The number of data slices is $min(\prod n_i, i \in \{1..m\}, D)$, where D is the number of data in the dataset.

The bug slices are slices with an accuracy lower than a given threshold, i.e., the model's overall accuracy. For each under-performing group, i.e., bug slices, the attribute choice can be used as the bug description (e.g., young people tend to have low accuracy).

**Bug classification.** To facilitate bug understanding, we categorize the bugs into two groups according to their root cause, namely rare case and spurious correlations. After classification, HiBug will give feedback to the user on the discovered bugs.

- **Rare case.** This type of bug often originates from insufficient training data [Eyuboglu et al., 2022]. To determine if a description is a rare case, HiBug first selects data from the validation set that matches the given description. Next, if these data have a high error rate and low frequency in the training set, we consider the bug a rare case bug.

- **Spurious correlation.** The training set we collect for model training can be biased. Therefore, the model might learn a dependence on some features that are not casual to the target. For example, in our lipstick-wearing classification task, the model learns to predict "not wearing lipstick" when the gender of people in the image is male. HiBug finds this correlation by calculating the conditional probability between the model's predictions and data description via the validation set. The correlation threshold $\alpha$ is a hyper-parameter.

## 3.3 Model repair

To improve the model, we propose a data selection approach and a data generation approach. For data selection, we first select data slices with the top error rate in the validation set. Then, we distribute the selection budget to the data slices based on the number of failures inside the slices. Finally, we select data matching the attribute values of the data slices from the unlabeled pool. The detailed algorithm is in the appendix. For data generation, we combine the class name with the attribute values of top error

Table 1: The percentage of problems that HiBug correctly identify the feature related to model's prediction and describe the correlation behavior of the model.

| Threshold | Number of Problems | Find Attr. | Find Des. |
|:---:|:---:|:---:|:---:|
| 0.7 | 457 | 84.9% | 72.6% |
| 0.8 | 352 | 77.8% | 64.2% |

rate data slices to create prompts. We then use these prompts as input for the generative model to generate new data. It is worth noting that the selected or generated data can also be a countermeasure for discovered bugs since attribute values related to bugs often appear in high error rate data slices.

## 4  Experiment

In this section, we validate our framework by (1) Can HiBug discover bugs? (2) What is the quality of discovered bugs? and (3) Can we improve the model with the discovered bugs? And How? We also present an ablation study to evaluate key factors of our method.

### 4.1  Can HiBug discover bugs?

**Experiment setup.** As discussed in section3.2, the model's failure on a data slice can be attributed to bugs such as insufficient training samples on rare cases or erroneous dependence on some feature. HiBug is the first framework that supports interpretable bug discovery. In this part, we evaluate our methods on the correlation discovery and rare case discovery tasks of the dcbench dataset proposed by domino [Eyuboglu et al., 2022].

The correlation discovery task in dcbench consists of 880 problems. Each problem contains common materials used in image classification, such as a model checkpoint, the model's predictions on a validation set, and labels for the validation set. Notably, each problem also includes a description of the erroneous correlation, indicating the name of the feature that the model's prediction is correlated with. For example, the prediction of a human-related classification model can be correlated with gender. The rare case discovery task in dcbench contains 118 problems. Apart from the common materials, each problem also has the name of a rare case.

**Correlation discovery.** The linear correlation coefficients between features and a model's predictions can differ in correlation discovery problems. Therefore, we evaluate HiBug using different correlation detection thresholds: 0.7 and 0.8. We only test on problems that have a correlation coefficient above the detection threshold $\alpha$. In our experiment, we evaluate whether HiBug can correctly identify the attribute value (feature) related to the model's prediction and describe the correlation behavior of the model. This is a challenging task, as HiBug not only needs to cover the feature in the attribute proposal and assignment process but also identify the correlation.

In Table1, we present two results of our methods. "Find Des." counts the cases where our method finds the exact name of a feature or a WordNet synonym [Miller, 1998]. "Find Attr." counts the cases where our method finds the correct feature or features in the same attribute. For example, when the model's prediction is correlated with "male", our method finds that the model's prediction is correlated with "female". "male" and "female" are both within the scope of the attribute "gender". It is worth noting that when the threshold is 0.7, HiBug accurately discovers a 72.6% correlation. This indicates that HiBug can effectively identify the model's erroneous correlation with the feature.

Previous methods have not explored interpretable bug discovery, so they cannot be used for direct comparison. Domino [Eyuboglu et al., 2022] is the state-of-the-art slice discovery method. As introduced in Section 2, it clusters data into slices and provides a description for each slice. Domino evaluates its clustering and describes abilities separately. In the correlation discovery task, Domino recovers ground truth bug slices with a mean precision-at-10 of 50%. It provides the correct description for a data slice 41% of the time. For bug identification, the probability that Domino provides correct feedback to the user can be the product of the above two probabilities. It is worth noting that Domino only produces a general description of the data in a bug slice, rather than identifying the bug itself.

Table 2: Basic information of three experiment settings. Split denotes the number of data in train set: validation set: test set: unlabeled pool.

| Dataset | Lipstick | RPC | ImageNet10 |
|---|---|---|---|
| Number of classes | 2 | 200 | 10 |
| Split | 80k:10k:10k:80k | 54k:25k:74k:49k | 10k:28k:2k:10k |

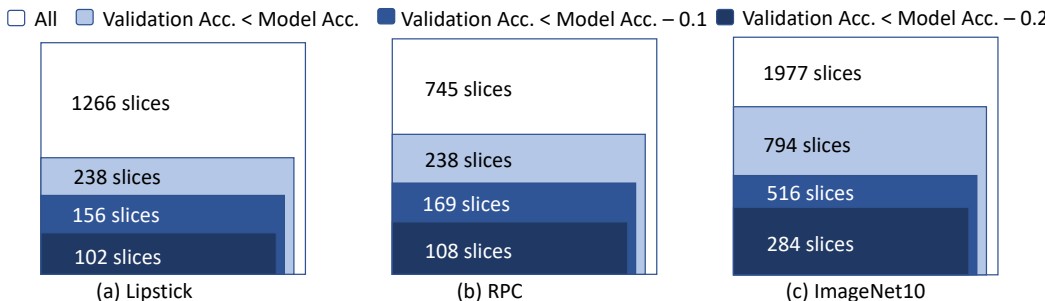

Figure 3: Number of bug slices uncovered by HiBug .

**Rare case discovery.** There are 118 problems in the rare case discovery task. In our experiment, we only use 99 problems for evaluation, as we filter out the problems where the given rare cases are not "cases" of the label. For example, in rare cases "menu" and "fare" for the label "food". In this experiment, we evaluate if HiBug can identify and describe the rare case. This task is also challenging since HiBug does not know the name of potential rare cases beforehand.

In 66.6% of the problems, HiBug find the name of the rare case or a WordNet synonym [Miller, 1998]. In 80.8% of the problems, HiBug finds the name or a sub-case of the target rare case. For example, HiBug finds "latte" but the name of the rare case is "coffee". For comparison, domino recovers ground truth bug slices with a mean precision-at-10 of 60% and gives the correct description for a data slice by 39%.

## 4.2 What is the quality of discovered bugs?

**Experiment setup.** HiBug produces data slices by grouping data with similar attribute values. We evaluate the discovered data slices of HiBug in three different settings. Basic experiment settings are present in Table2. We present the running outputs of HiBug in the appendix.

*Lipstick wearing classification* (lipstick for short): Firstly, we consider a setting where the model predicts incorrectly because of spurious correlation on the feature. We build a ResNet18 model to classify the usage of lipstick based on the CelebA [Liu et al., 2015] dataset. The reliance of features ("male" and "female") is injected by training with biased data. *Retail product classification* (rpc for short): We also consider a real-life classification task proposed in [Wei et al., 2019], which is a 200-classes classification on retail products. *ImageNet10 classification*: Finally, we apply our methods to a ten-classification task on ImageNet [Yang et al., 2022].

**Results.** We define data slices with low performance in the validation set as bug slices. Figure 3 shows the number of data slices under different validation accuracy thresholds. Although these slices are identified by the model's validation accuracy, their low-performing patterns are generalizable. To confirm this, we selected data that match the attribute values of the slices outside the validation and training sets (as shown in Table 3). We then verified the model's accuracy on this data. The results demonstrate that the model's accuracy on the bug slices is significantly lower than the overall accuracy. It is also worth noting that these data slices are coherent. A visualization of the data in a slice and the description of data in Figure4.

Table 3: This table shows the generalization ability of the error slices discovered by HiBug . For example, slices in column "Model Acc.- 0.1" has a validation accuracy lower than or equal to the original model's accuracy - 0.1. We select the data matching these descriptions of data slices from the data pool. The number is the model's accuracy on the selected data.

| Dataset | All slices | Model Acc. | Model Acc.- 0.1 | Model Acc.- 0.2 |
|---------|-----------|-----------|-----------------|-----------------|
| Lipstick | 0.907 | 0.842 | 0.819 | 0.804 |
| RPC | 0.778 | 0.668 | 0.590 | 0.539 |
| ImageNet10 | 0.873 | 0.849 | 0.823 | 0.774 |

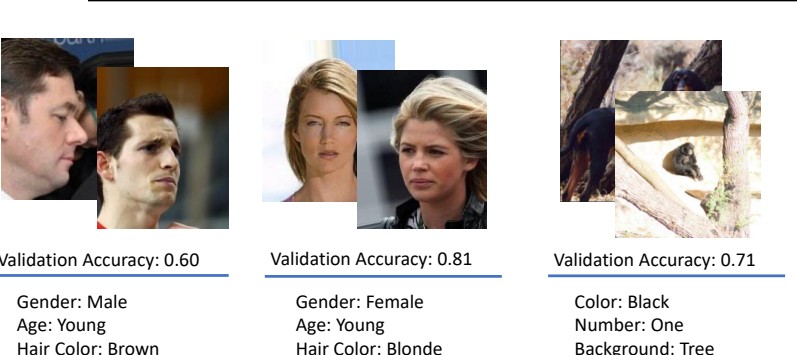

Validation Accuracy: 0.60

Gender: Male
Age: Young
Hair Color: Brown
Skin: White
Emotion: Pensive
Beard: No
Makeup: No

Validation Accuracy: 0.81

Gender: Female
Age: Young
Hair Color: Blonde
Skin: White
Emotion: Serious
Beard: No
Makeup: No

Validation Accuracy: 0.71

Color: Black
Number: One
Background: Tree
With Human: No
Coarse class : Animal

Validation Accuracy: 0.76

Color: White
Number: One
Background: Street
With Human: Yes
Coarse class : Vehicle

Figure 4: Visualization of data slices generated by HiBug .

### 4.3 Can we improve the model with the discovered bugs? And How?

**Experiment setup.** In continuation of the previous experiment, we extend our evaluation to ascertain whether the attribute values of bug slices can be further employed for data selection and data generation, ultimately contributing to model enhancement.

**Data selection.** Following the data selection method described in section3.3, HiBug selects a certain amount of data from the unlabeled data pool within a predetermined budget. The labels of this selected data are subsequently revealed, and the model is retrained with the original training set augmented by the selected data. We employ random selection and Corset [Sener and Savarese, 2017] — a representative active learning method — as baselines for comparison. The performance of each method is evaluated based on the error rate of the retrained models on the test set. We select 8,000, 500, and 1,000 new data points for the lipstick, RPC, and ImageNet10 tasks, respectively. As demonstrated in Table 4, our method consistently surpasses the baseline performance.

**Data generation.** The attribute values of bug slices produced by HiBug are represented in human-

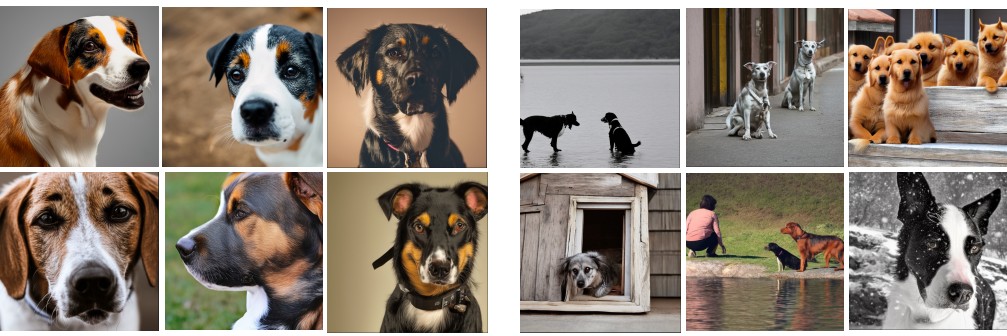

Figure 5: Image generated by class name(left) and images generated by prompts from HiBug (right).

Table 4: The model's error rate after retraining the original train set and data selected by different methods. We provide two results of the lipstick experiment – training from scratch or fine-tuning pretrained model. "Original" denotes the error rate of the model trained with original train set.

| Method | Original | Random | Coreset | HiBug |
|---|---|---|---|---|
| Lipstick scratch | 9.3% | 8.9% | 8.9% | 8.3% |
| Lipstick pretrain | 9.0% | 8.2% | 8.1% | 7.1% |
| RPC | 22.2% | 9.6% | 7.8% | 6.0% |
| ImageNet10 | 12.7% | 10.8% | 10.3% | 9.8% |

Table 5: Evaluation of generated data. "Class name" denotes data generated by prompts with class name only. "HiBug" represents data generated by prompts with class name and attribute values. In the first row, we present the original model's error rate on the generated data. In the second row, we present the model's error rate on test set after retraining with original train set and generated images.

| Number of images | 1000 | | 5000 | | 10000 | |
|---|---|---|---|---|---|---|
| Prompt | Class name | HiBug | Class name | HiBug | Class name | HiBug |
| Error rate on generated data | 16.5% | 37.2% | 16.6% | 35.9% | 16.6% | 35.5% |
| Retrain error rate on test set | 10.9% | 10.7% | 9.6% | 9.1% | 8.8% | 8.0% |

understandable terms. This facilitates downstream data generation for model enhancement, given that contemporary generative model, such as stable diffusion, also use language terms as input.

In our work, we create new data using prompts that encompass the class name (e.g., 'dog' or 'cat') and the attribute values of top low-performing data slices. For comparison, we also generate images based on prompts containing only the class name. This experiment is confined to ImageNet10 due to the limitations of the stable diffusion model in producing images satisfying the given class name in other settings.

Figure5 shows some examples of generated data. We observe that the images produced with prompts from HiBug exhibit greater diversity than those generated from class names alone. As indicated in Table5, when the model is tested on the generated data, the images created with a prompt from HiBug exhibit a higher error rate, suggesting that they represent corner cases for the model. Additionally, the model's performance improves when retrained with the combination of the generated data and the original training set. It's worth noting that a model trained with 10,000 generated data points outperforms a model trained with 1,000 selected data points from our data selection experiment. This finding indicates the potential of utilizing generated data for model improvement in future studies.

## 4.4 Ablation study

**Time cost.** In our experiments, we utilized an Intel(R) Xeon(R) Silver 4310 CPU @ 2.10GHz and an NVIDIA GeForce RTX 3090 GPU. The primary contributor to the computational cost in our framework, HiBug , is attribute assignment, which employs large vision-language models and scales linearly with the number of attributes. For a comprehensive study of the time cost of this component, we refer to [Li et al., 2022].

**Time complexity of bug discovery.** For the bug discovery of attribute combinations, we only focus on finding low-performance combinations. Its computational complexity is O(D), where D is the number of data in the dataset. Specifically, in the bug discovery process, every data has been assigned to an attribute combination. Firstly, we create a dictionary storing the number of correct data and the total number of data for every attribute combination. Then, we go through all the data in the dataset, collecting the data's correctness into the dictionary. Finally, we calculate the accuracy for each attribute combination in the dictionary while outputting those with low performance. Although the number of attribute combinations grows exponentially with the number of attributes, it is upper-bounded by the number of data in the dataset. Therefore, the time cost of this process can be viewed as 2D, which is O(D). In our experiment of bug discovery with 4536 combinations across a dataset of 20k images, the total execution time was a mere 1.06 seconds.

# 5 Conclusion

In summary, we present *HiBug*, an automated, interpretable model debugging framework that overcomes the limitations of traditional bug discovery methods by reducing human intervention and increasing bug coverage. HiBug employs pre-trained models like chatGPT to propose human-understandable attributes for specific computer vision tasks. Using vision-language models, it efficiently identifies shared visual attributes in underperforming data slices. This process aids in uncovering rare training data instances and spurious correlations, allowing for data-driven model enhancement. Experimental results underscore HiBug 's effectiveness, marking a significant advancement in interpretable model debugging.

Despite the progress made in our study, certain limitations persist. When the model's task is uncommon, the pre-trained language model may fail to yield relevant visual attributes. Furthermore, during the data generation phase, the generative model may encounter difficulties in creating images that align with the class name and our attribute values. Addressing these issues will be a core focus of our future work.

# 6 Discussion

**Impact of Components in HiBug.** In this paper, we choose BLIP and ChatGPT as the main components of HiBug. We acknowledge that while these large models have demonstrated strong ability, they can introduce bias. Please note that we rely on these models for attribute name and attribute value proposal and assignment. Even if they have some bias (e.g., they may neglect some kinds of attributes), we can still discover a lot of interpretable bugs (as evidenced by the experimental results). Data selection/generation can then help to repair the discovered bugs. However, due to the bias, some attributes remain uncovered and these bugs may not be found. Corresponding data cannot be generated for repair.

To address this problem, we provide a user interface that visualizes the entire process, which is presented in the Appendix. Users can make slight interventions to modify attributes and attribute values if necessary. Meanwhile, HiBug is highly flexible and modular. Therefore, designers can update any component in HiBug with a better counterpart.

**Social Impact.** In terms of potential applications and societal impacts, we want to emphasize that HiBug is rooted in industry engagement. We investigated the common debug flow in the industry and discussed it with a number of machine learning (ML) developers. The typical flow is to go through all failure cases and try to summarize failure patterns by ML developers themselves with ad-hoc solutions. After that, they collect new data related to failure patterns and re-train the model. This process requires intensive human efforts. Our original idea of HiBug is to build an AI assistant, like a co-pilot, to facilitate this process.

# Acknowledgments and Disclosure of Funding

This work is supported by the General Research Fund of Hong Kong Research Grants Council (Grant No. 14203521), CUHK SSFCRS Project 3136023, the National Natural Science Foundation of China (Grant No. 62306093), and the Guangdong Provincial Key Laboratory of Novel Security Intelligence Technologies (Grant No. 2022B1212010005).

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

# 7 Appendix

## 7.1 Model repair by HiBug

We elaborate on the details of data selection methods and data generation methods of *HiBug*.

**Data selection.** Our data selection method aims to identify crucial data for model repair from the unlabeled data pool. To achieve effective repair, we require the selected samples to exhibit both high error rates and diversity.

To accomplish this objective, our first step is to identify data slices with high error rates. Given a predetermined budget and an unlabeled data pool, we assign each data point in the unlabeled pool to its respective data slice based on attribute values. However, a challenge arises because the labels for these data points are unavailable. Therefore, we turn to the validation set, which already possesses known labels. By leveraging the labels in the validation set, we can calculate the failures and the error rate associated with each data slice. This information allows us to estimate the error rates of the unlabeled data slices.

Once we have obtained the error rate for each slice, we rank all data slices according to their error rates and select those with high error rates. This process is carried out iteratively, starting with the data slice that exhibits the highest error rate. We accumulate the number of failures associated with the selected slices as the iterations progress. The iteration continues until the accumulated number of failures equals or exceeds a threshold defined by the hyper-parameter $\beta$ multiplied by $D_F$. Here, $\beta$ represents the percentage of failures we aim to address, while $D_F$ corresponds to the total number of failures observed in the validation set.

After identifying slices with high error rates, we distribute the budget among the data slices proportionally to the number of failures they encompass. This allocation ensures that data slices with a higher number of failures receive a larger share of the budget, while also considering the diversity factor. Finally, we select unlabeled data from each data slice based on the budget allocated to that particular slice. During this selection process, we prioritize data points with lower confidence to maximize the potential for model improvement. In our experiments, we set the hyper-parameter $\beta$ to a fixed value of 0.9.

**Data generation.** *HiBug* can generate more related data (e.g., data with failure patterns) that can further improve the model performance. For data generation, we employ a straightforward approach. Firstly, we collect the attribute values of the top 200 data slices with the highest validation error rates for each label. Next, we construct a prompt that incorporates label name and attribute variables:

$$\textit{"A photo of *1 *2 *label *3, in a *4 background"} \tag{2}$$

During the data generation process, the variables *1, *2, *3, *4 are replaced by the values of attributes, such as "several" "black" "with a person" and "snow". The variable *label is substituted with the corresponding label name, such as "dog" or "cat."

We show the effectiveness of data generation with *HiBug* in ImageNet10. In this experiment, we generate a total of 10,000 data points, with 1,000 data points generated for each label. We down-sample the overall generated data to obtain results for 1,000 data points and 5,000 data points, as presented in Table 5. For comparison, we use a baseline method "Class name" that uses the following prompt, where the variable *label will be substituted with the corresponding label name during generation.

$$\textit{"A photo of *label."} \tag{3}$$

## 7.2 Introduction to the user interface

We offer a user interface for *HiBug* to facilitate more effective debugging. The user interface provides two primary functionalities. Firstly, users can easily view the distribution of attributes. For each attribute name, a histogram representing the various attribute values can be plotted, as demonstrated in Figure 6 (a). It's worth noting that we also provide additional flexibility by allowing users to input attribute names during the attribute proposal stage. This feature can be particularly useful when users

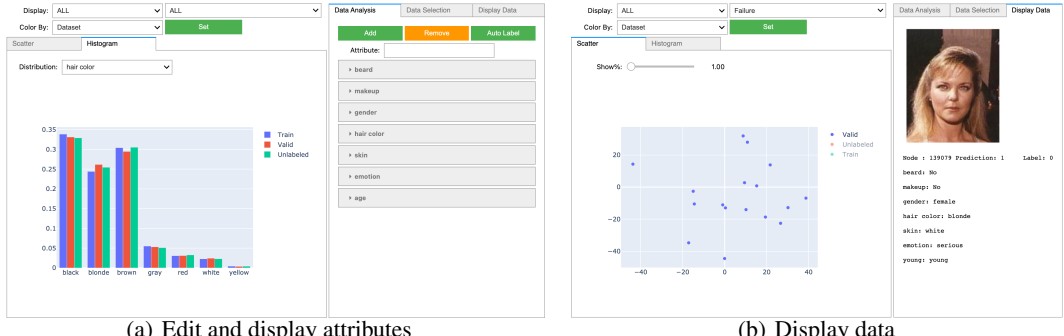

(a) Edit and display attributes          (b) Display data

Figure 6: In (a), we show that users can view the distribution of attributes by histogram (left tab), and propose or edit attributes (right tab). In (b), we show that users can view the distribution of features by scatter plot (left tab), and attributes of a specific data point in the scatter (right tab).

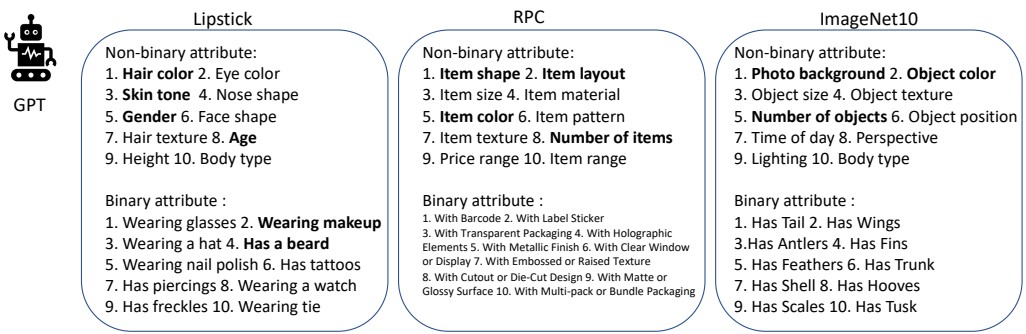

Figure 7: Attribute names proposed by chatGPT. The value of binary attributes are "yes" or "no".

are confident about specific tasks and seek to enhance the debugging process. Secondly, users can examine the data distributions in the embedding space. For example, we can display and analyze specific failures, as depicted in Figure 6 (b).

### 7.3 Experiment details

**The dcbench dataset.** HiBug aims to discover the spurious correlation and rare cases that exist in the models provided by the dcbench dataset. The correlation discovery task in dcbench consists of 880 problems. Each problem contains a model under test, the model's predictions on a validation set, and labels for the validation set. Each model contains some extent of spurious correlations, which are intentionally created for testing. For example, the necklace-wearing prediction of a model might correlate with gender. With HiBug, we can discover these spurious correlations and thus reduce the bias problems in machine learning models. The rare case discovery task in dcbench contains 118 models. Each model is a binary classification model. These models are created with intentionally inserted rare case problems. For instance, in the class of vehicles, they make the subclass carriages very rare, and the model prediction on carriages can be poor. Similarly, HiBug can detect these rare case problems and provide explanations.

To summarize, the dcbench contains many models with bias (cannot make proper predictions on some subset of data), and the goal of HiBug is to detect these problems, which can lead to a better machine learning model. We shall add a comprehensive description of the dataset in the revised version.

**ChatGPT attribute proposal.** Figure 7 shows the attribute names proposed by ChatGPT in the three experiment settings discussed in the main paper. We also provide additional flexibility by allowing users to select attribute names for HiBug to explore. We highlight the attributes selected for our experiments.

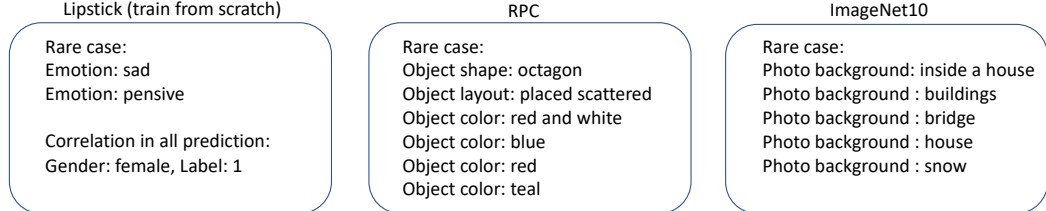

Figure 8: HiBug 's bug discovery outputs on three experiment settings.

Table 6: Given the same attribute list for each data, HiBug can discover more error slices with a larger validation set.

| Size of validation set | 15k | 10k | 5k | 1k | 0.5k |
|---|---|---|---|---|---|
| Number of error slices | 288 | 238 | 168 | 71 | 49 |

**HiBug 's bug discovery outputs.** Figure 8 showcases the bug discovery outputs obtained by HiBug in the three experiment settings outlined in the main paper. These outputs demonstrate the effectiveness of HiBug in identifying bugs and providing valuable insights to the user.

## 7.4 Further ablation studies

**Size of the validation set.**

We conducted additional ablation studies to explore the impact of the validation set size on HiBug 's performance. To quantify the error slices, we define them as data slices with validation performance lower than or equal to the overall validation performance. As presented in Table 6, we observed that HiBug can discover a greater number of error slices when the validation set size is larger. This phenomenon can be attributed to the larger dataset covering a broader range of attribute values, leading to the identification of a higher number of error slices by HiBug .

**Do bugs fixed after model repair?** In the lipstick experiment, the model's predictions are correlated with the gender of the person in the data. This correlation is detected in the bug discovery process of *HiBug*. For comparison, we use HiBug to test the model after repairing by data selection. The result is presented in Figure 9. We observe that most bugs discovered before repair disappear. This confirms again that our model repair strategy is effective.

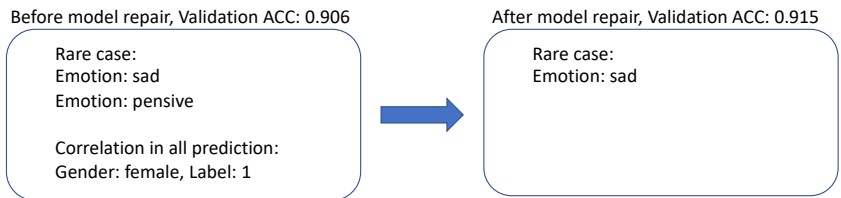

Figure 9: HiBug 's outputs before and after repair.

