# OpenReview forum: "HiBug: On Human-Interpretable Model Debug"
_NeurIPS.cc/2023/Conference — NeurIPS 2023 poster_

### Official Review · Reviewer_Zx32 · 2023-06-25

**Soundness:** 2 fair
**Presentation:** 2 fair
**Contribution:** 2 fair
**Rating:** 5
**Confidence:** 4

**Summary:**

This paper proposes a novel model-debugging method for deep learning-based classifiers. Technically, the proposed method first integrates a method for assigning attributes to training data. The method leverages a pre-trained large language model, such as chatGPT, to generate visual attributes based on task descriptions. By performing K-means clustering on the embeddings obtained from a pre-trained vision-language model called BLIP, It selects representative samples, i.e., centroid images, for each cluster. These samples are then queried with BLIP to obtain the specific attribute values. This process is applied iteratively to assign attribute values to all images in the dataset.
It then designs a method to discover samples that are likely to be misclassified by the model, so-called rare case bugs, and spurious correlation bugs. Rare case bugs aim to identify cases where the model's underperformance can be attributed to insufficient training data for a particular attribute value. The authors evaluate the validation accuracy for each group of data with a specific attribute value and flag attributes that exhibit a 0.1% or 0.2% drop in accuracy as rare cases. Spurious correlation bugs address situations where the model learns strong but incorrect associations among attributes. The authors measure linear correlations between attribute values generated by HiBug and compare them with ground truth data to identify such cases. The paper also proposes a model repair mechanism, which involves selecting unlabeled data with attribute values that have high validation error rates. Additionally, they combine class names with attribute values to create prompts for generating new data. Experimental results on benchmark datasets demonstrate HiBug's effectiveness in discovering rare cases and correlation errors. Furthermore, the paper shows the improvements achieved by the repaired model in terms of performance.



**Strengths:**

+ This paper studies an interesting problem, i.e., automatically identifying rare samples and spurious correlations that deep learning models may misclassify.

+ The proposed method leverages pretrained models to reduce human intervention.

**Weaknesses:**

First, the paper could provide a clearer distinction between the contributions and novelties of their work compared to the existing work, Domino [1].  Specifically, the rare case and spurious correlation are also discussed and identified in domino. For example, domino introduces artificial associations between features and uses their method to identify the introduced correlations, which is a similar task to HiBug. The key differences and challenges between HiBug and domino can be more clearly stated to stress the paper’s contributions. The difference in human interpretability between this work and domino can be more clearly stressed. In Section 4.1, the paper mentioned that HiBug can identify wrong correlations between features, and it still requires human experts to make the connections between such incorrect correlations and the model’s underperformance.  This step can also be performed on domino as a way to identify the bug in the model. The paper could provide further clarification on why and how HiBug has the ability to discover more interpretable and human-friendly bugs compared to other approaches.

Second, some of the definitions and statements are not clearly discussed in the paper. The motivation for choosing BLIP instead of other models that generate cross-modal representation is not clearly discussed. Although Figure 1 mentions that CLIP's embedding space is entangled and unsuitable for representative image selection, a motivation example illustrating the embedding space of BLIP would further clarify the motivation behind this choice. The computation process of linear correlation between the model’s predictions and data attribute is unclear. For example, in Section 4.1, the paper mentioned that the detection threshold for linear correlation is 0.7 and 0.8 without stating the computation method. In line 237, the paper claims that HiBug can discover the model's erroneous correlation with 72.6% accuracy, but the calculation of this accuracy measure is not clearly explained.

Another minor question: Table 3 reports the validation accuracy that is 0.1% smaller than the normal model validation performance. Is it computed on the mean of all data slices’ validation accuracy?

[1] Domino: Discovering Systematic Errors with Cross-Modal Embeddings, ICLR 2022




**Questions:**

Please see the weaknesses above.

**Limitations:**

The paper does not discuss the potential limitations, such as the generalizability and scalability of the proposed method. Another potential limitation might be its reliance on the performance of the pretrained models. The performance of the proposed method might be affected if the pretrained model cannot give accurate results.

---

> ### Author Rebuttal · Authors · 2023-08-09
>
> Thank you for pointing out our paper's problems. In the following parts, we address your concerns:
> 1. **First, the paper could provide a clearer distinction between the contributions and novelties of their work compared to the existing work, Domino [1].**
>     - Thank you for pointing out this problem. A comprehensive description of our contributions is in the top author rebuttal section. In general, HiBug has the following contributions compared to existing works like Domino: (1) HiBug provides a more interpretable and coherent data slice, (2) HiBug requires far less human effort, (3) HiBug generates a meaningful description of the data slice, and (4) HiBug can identify the type of bugs (e.g., rare case or spurious correlation). We will make it clearer in our revised paper.
> 2. **Some of the definitions and statements are not clearly discussed in the paper. The motivation for choosing BLIP instead of other models that generate cross-modal representation is not clearly discussed. Although Figure 1 mentions that CLIP's embedding space is entangled and unsuitable for representative image selection, a motivation example illustrating the embedding space of BLIP would further clarify the motivation behind this choice.**
>
>     - We would like to clarify that we do not use the BLIP as a substitute for CLIP to generate a better cross-modal representation.
>
>     - Previous works, such as Domino, use a pre-trained cross-modal model to extract image representations. Based on these representations, they cluster failures in the feature space and find corresponding text to explain the common features of the cluster. As they rely on the embedding space for clustering and explanation, the quality of the embedding space is important. However, the attributes in the embedding space are often entangled, leading to sub-optimal interpretation.
>
>     - In contrast, HiBug takes a completely different approach compared to existing methods (e.g., Domino) and does not rely on the embedding space for failure clustering. Instead, HiBug chooses to first group similar data together and then checks if they correspond to bug slices. In this process, BLIP is used for the VQA task, assigning each image with corresponding attribute values. This way, the slicing step is not affected by the quality of the embedding space.
>
> 3. **The computation process of linear correlation between the model’s predictions and data attribute is unclear. For example, in Section 4.1, the paper mentioned that the detection threshold for linear correlation is 0.7 and 0.8 without stating the computation method. In line 237, the paper claims that HiBug can discover the model's erroneous correlation with 72.6% accuracy, but the calculation of this accuracy measure is not clearly explained.**
>     - Computation process of linear correlation
>         - The linear correlation for a model's prediction $c_i$ and an attribute value $a_i$ is $P\left(c_i’ | a_i \right) $. We also refer to our common response (CQ2) for this question.
>     - Discover the erroneous correlation
>         - Each problem $p^i$ in problem set $P$ has a ground truth erroneous correlation ${c_i-a_i}$, which denotes a correlation between predicted label $c_i$ and attribute value $a_i$. HiBug can discover this correlation, if (1) $a_i$ is an attribute value in HiBug (2) the linear correlation of a_i and c_i exceeds the pre-defined threshold. We calculate the percentage that HiBug succeeds in overall problems.
>             - $\frac{|HiBug \ discover \ correlation|}{|P|}$
>     - We will add these formulas in the revised version.
> 4. **Another minor question: Table 3 reports the validation accuracy that is 0.1% smaller than the normal model validation performance. Is it computed on the mean of all data slices’ validation accuracy?**
>     - Yes

---

> > ### Comment · Reviewer_Zx32 · 2023-08-16
> >
> > Thank the author for the rebuttal.
> >
> > 1. **Clearer Distinction of Contributions:** The detailed description that the authors provided of HiBug’s contributions in comparison to Domino is helpful. The authors might want to elaborate more on these differences in a future version. For example, in what way does HiBug reduce human participation compared to Domino? A more formal description of the difference between the proposed method and existing methods is also appreciated (as promised by the authors).
> >
> > 2. **Motivation for Choosing BLIP:** The distinction that the authors made in how HiBug operates without relying on the embedding space for failure clustering provides a clearer understanding of the proposed method.
> >
> > 3. **Computation Process of Linear Correlation and Erroneous Correlation:** Thank you. This addresses my concern regarding linear correlation and erroneous correlation.
> >
> > I will update my score to positive.

---

> > > ### Author Response · Authors · 2023-08-17
> > >
> > > We thank the reviewer for the recognition of our work after the rebuttal. In the revised version, we shall thoroughly elaborate on our contributions compared to previous work (especially Domino).

---

### Official Review · Reviewer_48rC · 2023-07-06

**Soundness:** 3 good
**Presentation:** 4 excellent
**Contribution:** 3 good
**Rating:** 6
**Confidence:** 3

**Summary:**

HiBug seeks to identify (useful) NL descriptions of the "slices" of inputs on which some model (say an image classifier) has higher error rate, and perhaps even explain why the error rate is high (e.g., not enough training data in that space, or some data bias towards an unrelated correlation).

Prior work (saliency maps and Spotlight) identify latent-space regions where bugs are clustered, but humans need to come up with good NL summaries of those latent regions. AdaVision does iterative summarization, again with human assistance. Other lines of work use NL templates to produce summaries that match the buggy latent regions, but those templates sometimes come out incoherent.

The way HiBug does all this is as follows:
1. it identifies auxilliary attribute types that might be relevant to a task, e.g., "size" and "number of legs" are attribute types relevant to a classifier of images as dogs or not; it uses an LLM, prompted with the task (e.g., "does this image contain a dog?") and asked to identify what might be relevant attribute types (e.g., expecting an answer of "size, number of legs, hairiness"). So this step uses an LLM to guess interesting attribute "keys", relevant to a task at hand.
2. it labels dataset examples with their values for these task-specific auxilliary attribute types (they call this "attribute assignment", which basically means finding the value for each attribute), by using a vision LM (e.g., for image 1, size -> large, number of legs -> 3, hairness -> yes). It doesn't directly prompt the vision LM (visual question-answering model) to get the answers, although that would be trivial, because of high inference cost. Instead, it uses a cheaper, multi-step approximation:
    1. Uses a vision LM (BLIP) to map images to their embeddings, does K means clustering in embedding space, and selects centroids. The idea is that centroids will probably share attribute values.
    1. For each centroid example, it queries to VQA to get the actual (symbolic) values to the auxilliary attributes.
    1. It uses the values from all those cluster-examplar VQA answer to build a "vocabulary" of attribute values for each chosen auxilliary attribute.
    1. It embeds each attribute value (e.g., "size large" or "number of legs 3"), by itself, using the visual LM. This gives an embedding for every attribute value.
    1. Finally, for every dataset example (for which it already has an embedding), and for each auxilliary attribute, it checks the attribute-value embedding that's the most proximal to the example embedding. That enables a value assignment for each attribute for each example, without actually having to query the VQA model for each example.
3. It does inference on the initial task on each example (is it a dog?) and checks against golden labels (yes/no) to identify correct/incorrect predictions. With the correct/incorrect decision per example, and its auxilliary attribute value assignments, it can now slice the incorrect examples per single attribute value, or combinations of values for different attributes. If such a slice has a significantly higher error rate than the whole dataset, it's marked as a "buggy slice", and its corresponding attributes and values are the "description" for that buggy slice.
4. It draws an "explanation" for the buggy slice: if the training set slice has a small size, then the bugginess arises because the buggy set of attributes is rare; if there is a high linear correlation between the golden labels and the buggy slice of the validation data, then the dataset is biased.
5. Finally, to repair the model, HiBug does data selection, by choosing unlabelled data with buggy-slice attributes, labelling them, and augmenting the training dataset; it also does data generation, by creating an LLM prompt with the buggy-slice attributes, and generating synthetic examples.

Evaluation is done on the dcbench set from the Domino paper, which contains "buggy" model checkpoints, datasets, and the buggy correlation, or rare slice. HiBug is tested on identifying the slice description (either just the attribute "values", or also the attribute "keys"). Beyond identification of slices, HiBug was tested on 3 large tasks (e.g., ImageNet10), on which it identified buggy slices in validation data, which were then confirmed by looking at the performance of test data from those slices. Finally, model repair was evaluated and showed that performance can improve for biased models.


**Strengths:**

1. Except for task description, and data labelling during model repair, HiBug is fairly automatic and requires little human involvement, making the approach potentially impactful.
2. The performance on dcbench seems significant, for such a diverse dataset, compared to prior work (Domino), and while also offering qualitatively better feedback (the identification of the bug, rather than a description of a cluster).
3. Although the approach is fairly complex, it is described in sufficient detail to make reproducibility quite plausible. This is a well-written paper.
4. The approach seems fairly original (and non-trivial), and deviates from the style of prior art (e.g., clustering of failures in latent space). Also, it's quite creative to get around cost challenges.

**Weaknesses:**

1. Some assertions, especially in the design section, are provided without explanation or justification, making it hard to assess their veracity (see question 2).
2. I find Figure 5 a poor use of space. Showing images with increased diversity could be done any number of ways (e.g., randomly synthesizing prompts to an LLM) and doesn't prove anything. I'd move this text and figure to the appendix or remove.
3. I don't understand how you avoid bias inherent in the building blocks used in HiBug (e.g., BLIP or ChatGPT). Especially for model repair with data generation, bias seems problematic (question 1).
4. I wonder if the early parts of the approach (attribute discovery and value assignment) are a bit overengineered. Given that a human must write down a description for the task, couldn't the same human also describe some attributes that might be relevant, as well as the domain of values for each? Are the numbers of attributes and values that large that human enumeration is prohibitive? (question 3)

**Questions:**

1. Is using an LLM to do data generation prone to reinforcing bias? If the dataset in the LLM is already biased, how do you prevent model repair from reinforcing the same bias during data generation? [line 205]
2. How do you know which features are or are not causal to the target? [line 194]
3. How large is $|A|$ and $N$ typically, say for dcbench? Would it be practical for a human enumerate them manually, rather than using HiBench to discover them from LLMs/VQAs?

**Limitations:**

1. Since pre-existing, pre-trained models are used as building blocks to construct the domain in which buggy slices are identified, any bias in those building blocks might transfer to the results of the model. This limitation isn't mentioned.
2. A pre-trained vision LM isn't necessarily trained explicitly to separate any plausible attribute in its latent space. It's not clear where the approach of doing value assignment via the HiBug approach might fail in such cases.

---

> ### Author Rebuttal · Authors · 2023-08-09
>
> We sincerely thank your recognition of our paper. We address your concerns as follows:
> 1. **Some assertions, especially in the design section, are provided without explanation or justification, making it hard to assess their veracity.**
>     - We refer to the common response CQ1 to this question.
> 2.  **Find Figure 5 a poor use of space. Showing images with increased diversity could be done any number of ways (e.g., randomly synthesizing prompts to an LLM) and doesn't prove anything. I'd move this text and figure to the appendix or remove.**
>     - Thanks for pointing out this, we will move it to the appendix in the revised version.
> 3. **I don't understand how you avoid bias inherent in the building blocks used in HiBug (e.g., BLIP or ChatGPT). Especially for model repair with data generation, bias seems problematic. Is using an LLM to do data generation prone to reinforcing bias? If the dataset in the LLM is already biased, how do you prevent model repair from reinforcing the same bias during data generation?**
>     - We agree that while these large models have demonstrated strong ability, they can introduce bias. Please note that we rely on these models for attribute name and attribute value proposal and assignment. Even if they have some bias (e.g., they may neglect some kinds of attributes), we can still discover a lot of interpretable bugs (as evidenced by the experimental results). Data generation can then help to repair the discovered bugs. However, due to the bias, some attributes remain uncovered and these bugs may not be found. Corresponding data cannot be generated for repair. To address this problem, we provide a user interface that visualizes the entire process. Users can make slight interventions to modify attributes and attribute values if necessary.
> 4. **How do you know which features are or are not causal to the target?**
>     - We compute the linear correlation between the attribute value and the prediction label. Please refer to the common response (CQ2) for this question.
> 5. **I wonder if the early parts of the approach (attribute discovery and value assignment) are abit over-engineered. How large is |A| and N typically, say for dcbench? Would it be practical for a human enumerate them manually, rather than using HiBench to discover them from LLMs/VQAs?**
>     - In our experiments on dcbench, |A| is less than 10, and N is less than 50. The values in the experiment are relatively small and can be proposed by humans. However, this is not scalable. With HiBug, we can complete this task quickly and scale to more complex tasks.
>     - Specifically, HiBug can facilitate both the attribute discovery and value assignment processes. For attribute discovery, HiBug automatically proposes many attribute names and values. This is much faster than humans thinking of these words on their own, especially for complex tasks that require fine-grained descriptions of the images. These proposed names and values can also serve as inspiration for humans to choose from, thereby accelerating the attribute proposal step. For value assignment, HiBug automatically assigns attribute values to each image. Manually assigning attributes for each image is impractical, especially for modern tasks that often involve a vast amount of data.

---

> > ### Comment · Reviewer_48rC · 2023-08-18
> > **Response to rebuttal**
> >
> > Thank you for the detailed rebuttal.
> >
> > Response to #3: I didn't quite see how using the potentially biased models for value assignment isn't a problem. If, for example, a vision model decides that all people with long hair are female, and assigns the "female" value to the "gender" attribute for a bunch of images while trying to debug problems in the "is this a politician?" task, won't that lead to erroneous debugging results?
> >
> > I don't think that question is satisfactory, but it's still a hard problem with a good solution that improves automation, so I don't consider this a problem for this submission.
> >
> > Response to #5: I fully believe you that value assignment, done in an automated fashion, makes a lot of sense, and your engineering of that component seems sensible. I'm questioning whether using an LLM do discover the space of attributes and potential values is required, however. Perhaps an example in which the choice of attributes and their value domain aren't obvious and are error prone would help in the next iteration of the paper.

---

> > > ### Author Response · Authors · 2023-08-20
> > >
> > > We sincerely thank the reviewer for taking valuable time to respond to our rebuttal, and we try to address these issues as follows.
> > >
> > > 1. Response to #3: I didn't quite see how using the potentially biased models for value assignment isn't a problem. If, for example, a vision model decides that all people with long hair are female, and assigns the "female" value to the "gender" attribute for a bunch of images while trying to debug problems in the "is this a politician?" task, won't that lead to erroneous debugging results? I don't think that question is satisfactory, but it's still a hard problem with a good solution that improves automation, so I don't consider this a problem for this submission.
> > >
> > > - We agree with the reviewer that a biased pre-trained model could lead to debug failures with HiBug. For this specific "is this a politician?" task, suppose the low-performing slice for this task is  [“male”, ”long hair”], and the biased model assigns the female value to the "gender" attribute of all people with long hair, then HiBug would fail.
> > > - At the same time, we would like to point out that, the pre-trained large models show exceptional zero-shot capabilities and continue to improve over the years. They would still have bias, but as long as the bias is not so extreme, e.g., some long-haired males are not misclassified as females in this example, then HiBug would still be able to identify the low-performing slice.
> > > - Moreover, we have equipped HiBug with a user interface for attribute input and examination. For this particular example, users could choose to display samples with the "long hair" tag and possibly find the error.
> > >
> > > 2. Response to #5: I fully believe you that value assignment, done in an automated fashion, makes a lot of sense, and your engineering of that component seems sensible. I'm questioning whether using an LLM do discover the space of attributes and potential values is required, however. Perhaps an example in which the choice of attributes and their value domain aren't obvious and are error prone would help in the next iteration of the paper.
> > >
> > > We thank the reviewer for the recognition of our work. In our humble opinion, using LLM to discover the space of attributes and potential values is essential, because it is much easier for humans to choose from attributes and values proposed by LLMs than to think out all possible attributes and values on their own. Let us consider the rare case discovery experiment in Sec. 4.1. In this experiment, rare cases are often subclass of a labeled class. For example, carriage is a subclass of the vehicle class. If we use “subclass” as an attribute name, in order to discover potential value, we need to list a wide range of vehicle subclasses. It is likely to have missing items for humans without external help. At the same time, we acknowledge that LLMs are not omnipotent, and we suggest using HiBug as a Co-Pilot in debugging, together with the intellectual minds of humans.
> > >
> > > We shall add more discussions on the above issue in the revised version.

---

> > > > ### Comment · Reviewer_48rC · 2023-08-21
> > > > **Response v2**
> > > >
> > > > Thank you for your response to my response to #5. That makes sense. Some attributes are very vague, so deciding what constitutes potential values could be tricky to do. I will concede to your point here.

---

### Official Review · Reviewer_DbeT · 2023-07-09

**Soundness:** 3 good
**Presentation:** 3 good
**Contribution:** 3 good
**Rating:** 7
**Confidence:** 3

**Summary:**

The authors introduce HiBug, an approach to identify bugs in trained models in an interpretable fashion.  At its core, HiBug annotates inputs (images) with a set of attributes and values obtained using an LLM followed by a VQA step, and then identifies those combinations of attributes corresponding to subsets of test examples that underperform.  "Bugs" uncovered by this procedure are then fixed by requesting the label of either unlabeled instances or of synthetic instances, prioritized based on the distribution of buggy data subsets identified in the previous step.  HiBug is evaluated on a subset of the dcbench dataset (consisting of ~1000 ML problems and model checkpoints).

**Post-rebuttal update**: increased score by one point.

**Strengths:**

+ Tackles an important problem - semi-automated debugging of ML models.
+ Potentially very useful.
+ Text is easy to read.
+ Related work is well done.
+ Proposed approach is quite heuristic in nature, but overall sensible.
+ Proposed approach is quite straightforward, which is a plus.
+ Could serve as a basis for less straightforward techniques.
+ Experiments are quite extensive.

**Weaknesses:**

- Proposed approach is quite straightforward, which is also a minus, and specifically...
- ... I am on the fence regarding novelty.  While HiBug might be novel on paper, the real question is: how different is it from what developers debugging ML models already do on a day-to-day basis?
- Implicitly assumes LLM and VQA stages produce high-quality outputs.  This limitation is briefly mentioned in the conclusion, tho.
- No direct comparison with competitors - as there are none.  Not a major issue.

All in all, HiBug is not my cup of tea, but if it works and it is useful, I cannot really complain.

**Questions:**

Q1. "Recent research indicates that large language models, like ChatGPT, possess extensive general knowledge and are highly effective at answering questions" (p 4)  Could you please reference literature in support of this statement?

Q2. Spurious correlations: wouldn't it make more sense to compute correlations using a non-linear measure, like Spearman's rho?As I already mentioned, I am generally positive about the paper.

**Limitations:**

Briefly discussed in the conclusion.  The discussion is fair.

In addition to this, the authors could briefly comment on what they expect to happen when HiBug malfunctions - i.e., it reports non-existent bugs or neglects actual bugs - and how one should go about handling this.

---

> ### Author Rebuttal · Authors · 2023-08-09
>
> We sincerely thanks your recognition of our paper. We address your concerns as following:
> 1. **While HiBug might be novel on paper, how different is it from what developers debugging ML models already do on a day-to-day basis?**
>     - Before designing HiBug, we investigated the common debugging flow in the industry and discussed it with a few developers. Generally, ML developers manually go through all failure cases and attempt to summarize failure patterns themselves. To reduce the manual effort, existing works such as Domino [Eyuboglu et al., 2022] and Spotlight [d’Eon et al., 2022] try to cluster existing failures in the feature space and summarize common patterns. However, this flow—summarizing common patterns based on discovered failures—is problematic. For example, when the feature space contains entangled features or the discovered failures are sparsely distributed in the feature space, it becomes difficult to identify the true bug-related attributes.
>
>       Unlike existing efforts, HiBug follows a different debugging flow. We first group the data according to their attributes and then judge if there are underperforming ones. This way, underperforming slices must share a similar concept, leading to more accurate explanations. For a more detailed illustration, please refer to our clarification of novelty in the top section of the rebuttal.
>
> 2. **"Recent research indicates that large language models, like ChatGPT, possess extensive general knowledge and are highly effective at answering questions" (p 4) Could you please reference literature in support of this statement?**
>     - We shall add corresponding references in our revised paper. Some representative ones are:
>         - [1] Wei, J., Wang, X., Schuurmans, D., Bosma, M., Xia, F., Chi, E., ... & Zhou, D. (2022). Chain-of-thought prompting elicits reasoning in large language models. *Advances in Neural Information Processing Systems*, *35*, 24824-24837.
>         - [2] Touvron, H., Lavril, T., Izacard, G., Martinet, X., Lachaux, M. A., Lacroix, T., ... & Lample, G. (2023). Llama: Open and efficient foundation language models. *arXiv preprint arXiv:2302.13971*.
> 3. **Spurious correlations: wouldn't it make more sense to compute correlations using a non-linear measure, like Spearman's rho?**
>     - Please refer to the common response (CQ2) for this question. In general, non-linear correlations, such as Spearman correlation, might be inapplicable in some scenarios due to the model’s prediction confidence being inaccessible and untrustworthy. Currently, we use simple linear correlations to demonstrate the effectiveness of the HiBug. We acknowledge that other correlation coefficients may yield better results, and we intend to explore them in future endeavors.
> 4. **What do they expect to happen when HiBug malfunctions - i.e., it reports non-existent bugs or neglects actual bugs - and how one should go about handling this?**
>     - First, HiBug can hardly detect non-existent bugs. When identifying interpretable bugs, HiBug groups images with similar attributes into slices for performance evaluation. It's important to note that if the performance of a particular slice is lower than the average accuracy, we consider it a bug and take the corresponding attributes as the failure pattern. For HiBug to flag non-existent bugs, two requirements must be met: the slice must be underperforming, and the concept used for slicing must incorrectly characterize the data in the slice. These two requirements are unlikely to occur simultaneously. This is because, even if some data in the slices are incorrectly assigned attributes, it will not affect the bug discovery process (non-existent bugs will not be flagged). In the worst case, if most data in the slice is wrongly assigned the same attribute, they are likely to behave differently and are unlikely to cause a high level of misclassifications (resulting in an underperforming slice).
>     - Meanwhile, even if HiBug reports non-existent bugs, users can quickly identify the error. Firstly, bugs reported by HiBug are interpretable. Second, all bugs found by HiBug are presented in the user interface. Users can easily examine the data tagged with bugs to determine if the bug truly exists.
>     - The contribution of this paper is to automatically identify interpretable bugs, which could provide further repair suggestions for this model. We acknowledge that HiBug may overlook some interpretable bugs, especially when fine-grained attributes cannot be suggested by the attribute name and value proposal steps. However, with the development of vision-language models, we believe that the identification of more fine-grained attributes holds promise for discovering more bugs.

---

> > ### Comment · Reviewer_DbeT · 2023-08-21
> > **Reply**
> >
> > Thank you for your detailed reply, and apologies for not engaging in the discussion so far.
> >
> > 1. I see, this is sensible.  I will increase my score based on this.
> >
> > 2. Thank you.
> >
> > 3. Great, thank you.
> >
> > 4. Thank you for clarifying this point.  I still think it would make sense to briefly discuss issues possible malfunctions of HiBug in the paper.

---

### Official Review · Reviewer_GQqq · 2023-07-11

**Soundness:** 2 fair
**Presentation:** 2 fair
**Contribution:** 2 fair
**Rating:** 3
**Confidence:** 4

**Summary:**

 In this paper the authors describe  a system that introduces interpretability into the LLM model debugging. They use an LLM like chatGPT to  reveal interpretable  features from data for which the ML models don't perform well, With these features they find poorly performing data slices and provide identifiable attribute based or NL based description of visual features for these poor performing data slices. This, according to the authors, helps identify spurious correlations as well as rare (inadequate) datasets in training.

**Strengths:**

The biggest contribution of the paper is in attempting to bring in transparency and debugging where LLM models have poor performance. They rely on visual attributes and their value extraction from slices of data that show poor performance. Then map these to spurious corrleations or bias through rareness of certain attributes in the training set.

**Weaknesses:**

While the authors present a framework the biggest challenge is identifiying the attributes that are relevant to a certain task in the dataset .This is a manual process and can be challenging. The authors give one example of how it is done but that's not convincing enough that it can be easily done.
The second challenge is that they use another model to extract the attribute values of the identified attributes. It is not clear how effective this model is in identifying the values correctly.
The step of selecting representative images for BLIP they use k-means clustering. It is not clear how good these clusters are and what coverage and accuracy one gets. This seems like an important step in scaling the image selection
In the interpretable bug discovery step, every attribute, every combination of attributes have to be tried to group images and this step can be cumbersome and explosive.
In the bug classification step it is not clear how effective is the linear correlation step. No measurement described here.

**Questions:**

It will be great if the authors can address the questions in the weakness section.
1. How scalable is tha ttribute identifcation step based on tasks.
2. It would be good to address the goodness of the model that extracts the attribute values.
3. The representative selection step uses K-means clustering. How did you evaluate its goodness.
4. Please describe the complexity of the bug discovery step esp for combination of attributes.

**Limitations:**

It is not clear if the algorithm will help improve the bias in the training data by tihs technique. While the authors do address the problem of possible rarity of certain attributes in the training datatsets and spurious correlations to other attributes it is not clear if their proposed method solves that.

---

> ### Author Rebuttal · Authors · 2023-08-07
>
> Thank you for pointing out our paper's problems. We sincerely hope you can read our clarification of HiBug’s novelty in the author rebuttal section above. In the following parts, we address your concerns:
>
> 1. It will be great if the authors can address the questions in the weakness section.
>     1. **Goodness of ChatGPT, BLIP, K-Means.**
>         - We refer to the response to common question 1 for this question. Generally speaking, the main contribution of HiBug is proposing a novel and effective workflow for model debugging. We choose ChatGPT, BLIP and K-Means mainly because they are mature and common solutions for corresponding tasks. Our experiment evaluating the whole workflow has demonstrated the effectiveness of every component in HiBug. HiBug is highly flexible and modular. Therefore, we can update any component in HiBug with a better counterpart.
>         - We also provide the zero-shot classification of BLIP in some attributes of CelebA, where we have ground truth values for some attribute values. Gender classification: 99%, Age classification: 89%. For more evaluations, we refer to the original paper of BLIP[3].
>     2. **How scalable is the attribute identification step based on tasks?**
>         - The attribute identification step of HiBug is powered by ChatGPT, which owns a great knowledge base of reality. However, we acknowledge that ChatGPT can have some limitations in identifying attributes for uncommon tasks.
>         - To overcome this problem, we provide an easy-to-use user interface, which is shown in supplementary material, to help users go through failure cases of models and propose attributes by themselves. While manual attribute proposal still requires human efforts. We emphasize that when debugging a model, these efforts can be overlooked in comparison to the time saved by HiBug.
>         - Meanwhile, HiBug is a highly modular and flexible framework, the backbone GPT can be easily updated by any new language model or task-related knowledge graph.
>     3. **Please describe the complexity of the bug discovery step esp for combination of attributes.**
>         - Our rare case and correlation bug discovery is limited to one attribute value. For the bug discovery of attribute combinations, we only focus on finding low-performance combinations. As described in the ablation study, its computational complexity of it is O(D), where D is the number of data in the dataset. Specifically, in the bug discovery process, every data has been assigned to an attribute combination. Our bug discovery is as follows:
>             - We create a dictionary storing the correct number of data and the total number of data for every attribute combination.
>             - We go through all the data in the dataset, collecting data’s correctness to the dictionary.
>             - We calculate the accuracy for each attribute combination in the dictionary while outputting those with low performance.
>
>             Although the number of attribute combinations grows exponentially with the number of attributes, it is upper-bounded by the number of data in the dataset. Therefore, the time cost of this process can be viewed as 2D, which is O(D).
>
> 2. **In the bug classification step, it is not clear how effective is the linear correlation step. No measurement is described here.**
>     - We refer to our response to common question 2 for this question. In general, (1) we choose linear correlation because of its simplicity. Linear correlation is straightforward and easy to understand, and (2) non-linear correlation, such as Spearman correlation, might not be applicable in some scenarios. Since the model’s prediction confidence can be inaccessible and untrustworthy. We will implement other correlation coefficients in HiBug as optional choices.
> 3. **While the authors do address the problem of possible rarity of certain attributes in the training datasets and spurious correlations to other attributes it is not clear if their proposed method solves that.**
>     - This is an insightful question. In the data selection and data generation process, HiBug finds attribute combinations that have low performance and selects or generates data with those attribute values for model training. This process inherently can fix the bug found by HiBug. For example, we have attributes [“gender”, “Age”] and labels {0, 1}.
>         - If “male” is a low-performing rare case. Attribute combinations [ ”male”,  x] will naturally become low-performing slices, where x can be { ”Young”, “Middle-Age”, “Old”}. Then in the data selection process, new data of “male” will be selected. Then the rare case problem can be alleviated.
>         - If “male” is correlated with label 0. Then in the data generation process, where the label can be used as an attribute. [“male”, x, 1] will become low-performing slices and corresponding data will be generated. Then the model’s dependence on “male” is reduced.
>     - We also present the bug discovery results before and after fixing in our paper appendix. Bugs disappear after fixing by data selection of HiBug. Moreover, our experiment has shown that HiBug can ensure a better model's performance by data selection and data generation compared with baselines.

---

> > ### Author Response · Authors · 2023-08-18
> >
> > Dear Reviewer GQqq,
> >
> > Thanks again for your insightful questions. We hope we have addressed your concerns in our rebuttal. Specifically, we address your concerns regarding the goodness of the various components in HiBug in the top rebuttal section. We have also provided detailed answers to the scalability, time complexity, and effectiveness of HiBug's workflow.
> >
> > Considering the deadline for the discussion period is approaching, we would like to know whether you are satisfied with our rebuttal. We appreciate your time and effort very much and we would be happy to provide further clarifications, if any.
> >
> > \- Submission14450 Authors

---

### Author Rebuttal · Authors · 2023-08-07

We sincerely thank all reviewers for their constructive comments and recognition of our work's strength.

Before addressing common questions, we clarify the novelty of our paper as follows:
### Novelty
- The problem HiBug focuses on is critical yet under-explored. Before designing HiBug, we investigated the common debug flow in the industry and discussed it with a few developers. The typical flow is to go through all failure cases and try to summarize failure patterns by ML developers with ad-hoc solutions. After that, they collect new data related to failure patterns and re-train the model. This process requires intensive human efforts. Our original idea of HiBug is to build an AI assistant, like a co-pilot, to facilitate this process.
- The main contribution of HiBug is proposing a useful and novel workflow that automates the process of data/model debugging and provides interpretable feedback. Previous attempts in this direction are either not interpretable or require intensive human effort, as discussed in the related work section. HiBug’s novel workflow can overcome these problems.

Thanks to Reviewer Zx32’s constructive suggestion, we elaborate on the difference between HiBug and Domino, which is a representative work of previous solutions. In short, Domino clusters failure cases of a model in the CLIP embedding space and find a description for the cluster from a pre-designed large corpus, while HiBug finds attributes and attribute values for data, and uses attribute values to cluster data and discover bugs. Specifically,

    1. HiBug identifies low-performing data slices in a more coherent way. Firstly, Domino might produce semantically incoherent data slices. For example, in Figure 1 of our paper, data with similar attributes are not close in the embedding space. This phenomenon is also observed by [1]. HiBug addresses this issue by directly clustering data with attribute values, inherently ensuring the coherence of data in a cluster.

    2. HiBug requires far less human effort. Although Domino can automatically find a description for data slice. It is worth noting that the description comes from a human-designed corpus. In the appendix of Domino's paper, they introduce how to build the task-related corpus, which includes human proposal and programmatic collection from Wikipedia. In contrast, HiBug does not require a corpus and identifies attributes directly from the dataset. This is powered by recent large language models, and it only requires a description of the task.

    3. HiBug generates a meaningful description of the data slice. Since Domino generates descriptions by combining words in the corpus, it often produces a sentence that contains irrelevant or contradicting words, such as “a photo of setup by banana” and “a photo of skiing at sandal”. HiBug avoids this issue in two ways. Firstly, data can only have one value from the same attribute. For example, it can only get one value from set {”male”, “female”}. Therefore, contradicting words cannot appear in HiBug. Second, attribute values are collected from the dataset by LLM and vision-language models. Therefore, only words relevant to the data can appear.

    4. HiBug can identify bugs more effectively. A description of a data slice is different from a description of a bug. Domino only finds and describes the low-performance data slices. It does not explicitly tell what kind of bug the data slice is related to. Therefore, human experts need to go through all data in the slice to find if it is a rare case or a spurious correlation. In contrast, HiBug can explicitly show the bugs.
- **This paper focuses on illustrating the novel debug workflow of HiBug. Therefore, we do not provide a detailed evaluation for each component in HiBug.** We select methods such as ChatGPT and BLIP mainly because they are easy to use and provide good solutions. Our experiment evaluating the whole workflow has demonstrated the effectiveness of every component in HiBug. We also would to point out that HiBug is highly flexible and modular. Therefore, designers can update any components in HiBug with a better counterpart, if needed.

### Common Questions

1. **Lack of explanation or evaluation of components in HiBug, such as choice of LLM, vision-language model, and clustering algorithm.**
    - Please refer to the last paragraph in the novelty clarification for this section.
    - Since Reviewer GZbC and two Ethic Reviewers also mention the performance evaluation of BLIP. We provide the zero-shot classification of BLIP in some attributes of CelebA, where we have ground truth values for some attribute values. Gender classification: 99%, Age classification: 89%. For more evaluations, we refer to the original paper of BLIP.
2. **Why use linear correlation? / How linear correlation is computed? / Why not use other correlation coefficients?**
    - Linear correlation is the same as the conditional probability in our paper. It computes that for data with attribute value $a_i$, the probability that the model predicts label $c_i$. The formula simply goes as：
        - $P\left(c_i | a_i \right) $
    - We choose linear correlation because of its simplicity, and the results show its effectiveness.
    - Non-linear correlation, such as Spearman correlation, might be inapplicable in some scenarios. For a multi-classification task, one way to apply non-linear correlation is by computing the correlation between the confidence of a label in the model’s prediction and the appearance of an attribute value. However, (1) the model’s prediction confidence can be inaccessible, and (2) some previous studies have shown that model’s over-confident nature makes the confidence value untrustworthy.  In contrast, linear correlation only requires the model’s prediction results and is always credible.
    - Nevertheless, we plan to implement other correlation coefficients in HiBug as optional choices.

[1] Adaptive Testing of Computer Vision Models, arXiv.

---

### Decision · Program_Chairs · 2023-09-21

**Decision:**

Accept (poster)

**Comment:**

The paper received largely positive reviews, and the rebuttal helped clarify some of the remaining points. One reviewer noted a few weaknesses, and the authors had strong responses to support the paper. LLMs are not perfect for selecting the attributes, but overall the paper is interesting as noted by other reviewers, and this aspect could be refined by follow-up work in the future.